# Nitrate Sensor with a Wide Detection Range and High Stability Based on a Cu-Modified Boron-Doped Diamond Electrode

**DOI:** 10.3390/mi15040487

**Published:** 2024-04-01

**Authors:** Shengnan Wei, Danlin Xiao, Yang Li, Chao Bian

**Affiliations:** 1State Key Laboratory of Transducer Technology, Aerospace Information Research Institute, Chinese Academy of Sciences, Beijing 100190, China; weishengnan11@163.com (S.W.); xiaodanlin1998@163.com (D.X.); 2School of Electronic, Electrical and Communication Engineering, University of Chinese Academy of Sciences, Beijing 100049, China

**Keywords:** electrochemical sensor, boron-doped diamond electrode, nitrate determination, Cu-modified, water pollution monitoring

## Abstract

This paper describes an electrochemical sensor based on a Cu-modified boron-doped diamond (BDD) electrode for the detection of nitrate-contaminated water. The sensor utilizes the catalytic effect of copper on nitrate and the stability of the BDD electrode. By optimizing the electrolyte system, the linear detection range was expanded, allowing the sensor to detect highly concentrated nitrate samples up to 100 mg/L with a low detection limit of 0.065 mg/L. Additionally, the stability of the sensor was improved. The relative standard deviation of the current responses during 25 consecutive tests was only 1.03%. The wide detection range and high stability of the sensor makes it suitable for field applications and the on-site monitoring of nitrate-contaminated waters.

## 1. Introduction

Water is essential for life, but the pollution of the limited freshwater resources has become a major concern due to the large population growth, rapid industrialization and urbanization, and reckless exploitation of natural water bodies [1]. Nitrate is a major pollutant of global surface water and groundwater [2]. Improper disposal of livestock waste, excessive use of agricultural fertilizers, and wastewater discharge have exacerbated nitrate pollution in the aquatic environment, posing a threat to the safety of drinking water for humans [3]. Studies have shown that although nitrate intake through drinking water accounts for only 5% of the total daily intake, it is more likely to cause health problems than nitrate ingestion through food intake [4]. Excessive nitrate in the human body can increase the risk of cancer and may also lead to diabetes, miscarriage, thyroid disease, blue baby syndrome, and methemoglobinemia [5,6]. The World Health Organization has set a standard for nitrate in drinking water of 50 mg/L (NO^3−^-N approximately 11.3 mg/L) [7,8], while the Chinese Surface Water Quality Environmental Standard sets an upper limit of 10 mg/L (NO^3−^-N) for nitrate concentration in drinking water [9]. Countries such as the United States, Canada, Australia, and the European Union have already established monitoring and management regulations or legislations targeting nitrate pollution [10]. Currently, approximately 7.83% of China’s major rivers have nitrate concentrations that exceed the national drinking water standard of 45 mg/L [11]. Nitrate concentrations in rivers such as the Mudan River (Linkou County), Haihe River (Beijing), and the Yangtze River estuary (Shanghai) even exceed 90 mg/L, indicating severe pollution [11]. Therefore, it is urgent to monitor and treat nitrate-polluted waters in real time to prevent further pollution.

The existing nitrate detection methods mainly include optical methods and electrochemical methods. Although optical methods have a low lower limit of detection and high accuracy, they require the preparation of complex detection substrates to bind the target [12]. Optical methods, such as chromatography, colorimetry, and spectrometry, require the regular collection of water samples for laboratory testing. These methods also require expensive laboratory equipment, complex chemical reagents, and professional operators, which increases the cost of detection and generates more chemical waste [13]. In recent years, studies have focused on developing portable optical nitrate detection systems. For example, research has been conducted on spectrophotometric reagentless testing [14] and the creation of an automated system for spectrophotometric nitrate detection [15]. Although these methods solve the problems of laboratory testing and reagents, their linear detection range remains narrow, and their upper limit of detection is too low for detecting high concentrations in polluted waters. To address these issues, alternative methods that are more cost-effective and environmentally friendly should be explored. Electrochemical methods are a suitable choice for on-site nitrate detection due to their high accuracy, wide measurement range, simplicity, and low cost. Electrochemical methods involve a charge transfer process on the electrode surface, resulting in a change in potential or current. This change is directly proportional to the concentration of the measured substance, allowing for the calculation of the target analyte’s concentration [16]. The electrochemical detection methods mainly include amperometric methods and potentiometric methods. The potentiometric method involves a two-electrode system comprising an ion-selective electrode (ISE) and a reference electrode (RE) to analyze and calculate the concentration of the target ion. This is achieved by measuring the potential difference between the two electrodes. The ISE surface is an ion-selective membrane (ISM) with an ion-selective carrier that permits only the target ions to be detected to pass through [17]. The potentiometric methods have limited anti-interference ability and a short service life due to their ion-selective membranes. This makes them unsuitable for long-term accurate detection in complex water samples in actual scenarios [18]. The amperometric method is based on a three-electrode system with a working electrode (WE), a reference electrode (RE), and a counter electrode (CE) [16]. Detection is mainly achieved by determining the current generated by the oxidation or reduction of the measured substance at the working electrode [19]. The amperometric method primarily employs the catalytic reduction reaction of nitrate by various metals. Copper is currently the most promising catalytic metal due to its ability to limit hydrogen adsorption on the electrode surface and promote nitrate adsorption sequentially [20]. To improve the performance of copper in nitrate detection, many copper-based electrodes have been studied and used. For example, Li, Y. et al. [21] deposited nano-copper clusters on platinum microelectrodes to increase the effective catalytic area of copper and improve the sensitivity of nitrate detection. Li, G. et al. [22] designed a novel copper/carbon nanofiber/carbon fiber microelectrode (Cu/CNF/CFE) and detected nitrate in a simple, inexpensive, and convenient manner. Other examples include copper nanowire arrays [23], oxide defect (OD) Pt-Cu electrodes [24], and copper nanowire networks [25]. However, many detection methods prioritize sensitivity over practical application, neglecting the fact that water samples may contain nitrate concentrations that span a wide range of values. Additionally, some electrodes require delicate and complex modification processes that prevent their reuse. Boron-doped diamond (BDD) is a promising material for electrochemical sensing applications due to its low background current, wide potential window, and superior antifouling properties compared to other carbon-based electrodes [26]. BDD exhibits excellent stability due to its chemical inertness, high hardness, and thermal conductivity, which are similar to those of diamond. This makes BDD electrodes suitable for use in corrosive solutions without experiencing any microstructural or morphological degradation. Moreover, BDD has high hydrogen and oxygen evolution potentials; so, the surface of BDD will not react with pollutants and impurities causing reduction or oxide fouling due to the precipitation of hydrogen or oxygen during the detection process [27]. BDD electrodes have the dual function of pollutant detection and removal [26]. Hydroxyl radicals (OH^−^) can be generated on BDD electrodes to degrade organic micropollutants by applying high voltage [28], demonstrating the high stability of BDD electrodes.

This study combines the stability of BDD with the catalytic effect of Cu on nitrate to construct a nitrate detection sensor based on a Cu-modified boron-doped diamond (BDD) electrode. Through the optimization of the detection electrolyte solution environment, the detection range of the sensor was broadened, and high stability was obtained, enabling the sensor to detect high concentrations of nitrate and achieve the on-site monitoring of nitrate-contaminated water areas. 

## 2. Materials and Methods

### 2.1. Instruments and Reagents

We used acetone, ethanol, H_2_SO_4_ (Beijing Chemical Factory, Beijing, China), KH_2_PO_4_, CuSO_4_·5H_2_O, Na_2_SO_4_, NaNO_3_, CH_3_COONa, NaCl, Na_2_SiO_3_, and Na_2_CO_3_ (China National Pharmaceutical Group Chemical Reagent Co., Ltd., Shanghai, China); all reagents used were of analytical grade.

We used the Reference 600 electrochemical workstation (Gamry Instruments, Warminster, PA, USA); the S-480 field-emission scanning electron microscope (SEM, Hitachi, Japan); the PHS-3C pH meter (Shanghai Leici Instruments Co., Ltd., Shanghai, China); and the Millipore Direct-Q deionized water system (Merck Millipore, MA, USA).

### 2.2. Fabrication and Pretreatment of the Electrodes

Boron-doped diamond (BDD) film electrodes were prepared on a silicon wafer by chemical vapor deposition [29] and diced, obtaining a diameter of 3 mm. Then, the electrodes were cleaned by sequential ultrasonic treatment in acetone, ethanol, and deionized water for 5 min each before use. After that, the electrodes were subjected to a +3 V voltage for 120 s in 0.5 mol/L H_2_SO_4_ to remove organic contaminants on their surface. Then, cyclic voltammetric scanning in the range from −3 V to +3 V at a scan rate of 50 mV/s was performed to activate the electrodes. Copper as the sensing material for nitrate was modified on the electrode by using the cyclic voltammetric method. The cyclic voltammetric method comprised 10 cycles of scanning from 0 V to −0.8 V at a scan rate of 50 mV/s. The deposition solution was a 0.15 mol/L CuSO_4_ solution at pH 1.

### 2.3. Determination of Nitrate

As shown in Figure 1, the electrochemical sensor developed for nitrate determination is a three-electrode system, with a BDD electrode modified with Cu as the working electrode, a bare BDD electrode as the counter electrode, and a Ag/AgCl electrode as the reference electrode. The Cu-modified BDD electrode surface can be renewed by applying a positive voltage, and the electrodes can be re-modified and reused again. The concentration of Na_2_SO_4_ and the pH of the electrolyte solution were optimized. For actual water samples detection, H_2_SO_4_ and Na_2_SO_4_ were added before detection to establish a suitable electrolyte environment. Linear sweep voltammetry was used for nitrate detection, with a scan voltage range of −0.8 V to −0.1 V and a scan rate of 50 mV/s. The reduction peak current was recorded to determine the concentration of nitrate.

## 3. Results and Discussion

### 3.1. Principle of Nitrate Detection

In acidic environment, copper can catalyze the complex reduction reaction of nitrate ions. The reaction produces nitrite, ammonium ions, and ammonia, among other products. However, most of these products are only intermediate products of the reaction. Ammonia ions are widely recognized as the final product of the reaction [30], as shown in Figure 2 and Equation (1). The nitrate reduction current response in this reaction is proportional to the nitrate concentration. Therefore, the nitrate sensor, based on copper-modified BDD in this study, calculates the nitrate concentration by analyzing the magnitude of the nitrate reduction peak during the reaction.
NO_3_^−^ + 10H^+^ + 8e^−^ → 3H_2_O + NH_4_^+^,(1)

### 3.2. Characteristics of the BDD Electrode

The electrochemical properties of the BDD electrode were characterized in the potassium ferricyanide system. Figure 3a displays the measured cyclic voltammetry (CV) curves of the BDD electrode in a 5 mM Fe(CN)_6_^3−/4−^ potassium ferricyanide system at different scan rates (10 mV/s, 20 mV/s, 50 mV/s, 100 mV/s, 200 mV/s, and 500 mV/s). Each CV curve exhibits a pair of redox peaks, corresponding to the redox processes of [Fe(CN)_6_]^4−^/[Fe(CN)_6_]^3−^ ions. The peak at a higher potential represents the oxidation peak, while the peak at a lower potential corresponds to the reduction peak. The oxidation and reduction peaks of the CV curves basically show a mirror-like symmetry, and the peak currents are approximately equal. As the scan rate increased, the peak values of both oxidation and reduction peaks also increased. Simultaneously, the oxidation peak potential shifted towards a positive direction, while the reduction peak potential moved towards a negative direction. In summary, the CV curves indicated that the [Fe(CN)_6_]^4−^/[Fe(CN)_6_]^3−^ couple reaction on the BDD electrode was a quasi-reversible reaction. Therefore, the BDD electrode can serve as a reactive working electrode for the electrochemical detection of redox reactions.

Figure 3b shows the relationship between the oxidation peak current and the square root of the scan rate at different scan rates. The redox peak current of the BDD electrode showed a linear relationship with the square root of the scan rate, indicating linear ion diffusion from high to low concentrations in the scanned solution. Because ion diffusion primarily controls electrochemical reactions, the reaction of the BDD electrode as a working electrode can be considered an electrochemical reaction.

The surface morphology of the BDD electrode and its morphology after copper deposition via cyclic voltammetry are presented in Figure 4. As shown in Figure 4a–c, the surface of the bare BDD electrode was dense and granular. This provided the growth points for copper deposition, resulting in a more reliable modification of the electrode surface by copper. Figure 4d,e shows the copper clusters deposited by cyclic voltammetry. Compared to the bare BDD electrode surface, the copper clusters exhibited porous and dendritic structures, significantly increasing the sensitive area of the electrode. This provided more catalytic reaction sites and enhanced the catalytic performance of the sensor. Furthermore, BDD electrodes can be consistently modified and utilized by applying a positive voltage to eliminate the modified copper and other contaminants from the electrode surface. This prevents the inconsistency that arises when an electrode’s catalytic performance decreases during the detection process, resulting in steady detection sensitivity and improved electrode stability. 

### 3.3. Optimization of the Experimental Parameters for Nitrate Detection

#### 3.3.1. Effect of Na_2_SO_4_ Concentration

The addition of Na_2_SO_4_ to the nitrate detection solution increased the ionic strength of the solution, enhanced the reduction current, and improved the detection sensitivity. Figure 5 shows the current responses of the same nitrate concentration under different Na_2_SO_4_ concentrations. When the concentration of Na_2_SO_4_ was less than 100 mM, the reduction current response increased with the increase in Na_2_SO_4_ concentration and reached a maximum at 100 mM. When the Na_2_SO_4_ concentration was greater than 100 mM, the reduction current decreased with the increase in Na_2_SO_4_ concentration. The reason for this is that the higher concentration of electrolytes in the solution caused an increase in conductivity, which in turn led to an increase in background noise and a decrease in the reduction current signal. Therefore, the optimal Na_2_SO_4_ concentration in the nitrate detection solution was 100 mM.

#### 3.3.2. Effect of Substrate pH

During nitrate detection, the electrocatalytic effect will decay along with the process of copper oxidation on the electrode surface [31]. Maintaining an acidic pH in the nitrate detection solution could prevent the oxidation of copper on the electrode while promoting the reduction of nitrate. However, excessive hydrogen ions can easily lead to hydrogen evolution during the reaction. To determine the optimal pH value of the detection solution, this study tested the current response of the electrode to the same concentration of nitrate under varying pH conditions (Figure 6a). At a pH of 1 or lower, the high concentration of hydrogen ions caused significant hydrogen evolution reactions, which could obscure the nitrate current response with the hydrogen ion reduction peak. Additionally, the copper film on the electrode surface was susceptible to damage and peeling. As the pH increased above 1, the current response decreased due to the weakened positive incentive for the nitrate reduction reaction caused by the decrease in hydrogen ion concentration. Additionally, the weaker acidic environment reduced the inhibition of copper oxidation. Figure 6c shows the concentration gradient linear-scan curve when the pH of the nitrate detection solution was 1.5, and Figure 6d shows the concentration gradient linear-scan curve when the pH was 2. By comparison, it is evident that the upper detection limit of nitrate at pH 2 was significantly lower than that at pH 1.5. Figure 6b presents the linear fitting curves of the concentration gradient from 0 to 30 mg/L for the nitrate detection solutions at pH = 1.5 and pH = 2. At pH = 2, the nitrate current response saturated when the nitrate concentration exceeded 15 mg/L, resulting in decreased detection sensitivity. Additionally, the fitting curve’s overall linearity and sensitivity at pH = 2 were lower than those at pH = 1.5. Therefore, increasing the pH also decreased sensitivity, linearity, and the detection upper limit.

### 3.4. Performance for Nitrate Determination

Subsequently, the response current values of the sensors were examined in relation to the concentration of nitrate ions using linear scanning. As demonstrated in Figure 7, the peak sensor response current was directly proportional to the concentration of nitrate ions. The sensor was able to detect nitrate in a wide concentration range, with segmented linear relationships in the ranges of 0.07–3 mg/L (Figure 7a) and 3–100 mg/L (Figure 7b). And the sensitivities were 3.5 µA·mg·L^−1^ and 5.33 µA·mg·L^−1^, respectively. The detection limit for nitrate was calculated to be 0.065 mg/L by dividing the three-time standard deviation of the blank signal by the slope of the calibration curve.

### 3.5. Anti-Interference Ability, Selectivity, and Repeatability

To achieve real-time nitrate detection on-site, a sensor should have good anti-interference ability and selectivity. Therefore, experiments on anti-interference ability and selectivity were conducted on this developed nitrate sensor. CH_3_COO^−^, Cl^−^, CO_3_^2−^, PO_4_^3−^, SO_4_^2−^, SiO_3_^2−^ at 10-fold concentrations were added to a 1 mg/L nitrate solution. The nitrate sensor developed in this study was used to conduct three repeated measurements on the solutions without and with the added 10-fold concentrated interfering ions. The anti-interference test results are shown in Figure 8a. The interference ions had a relatively small impact on the sensor, with current response deviations of less than 6%. Figure 8b presents the selectivity test results of the sensor. The response current of the sensor to other anions at a 10-fold concentration was less than 3%. These results indicate that the sensor exhibited excellent anti-interference ability and selectivity.

Since the detection target of the sensor in this study was high-concentration nitrate-polluted water bodies, further anti-interference and selectivity tests were conducted in high-concentration nitrate environments. Figure 8c,d reports the anti-interference test and selectivity test results using 10 mg/L nitrate solutions in the presence of other anions at a 10-fold concentration. Under the interference of ions, the deviation of the current response of the nitrate sensor to nitrate was less than 9%. Although this deviation was larger than that under low anion interference, it still demonstrated good anti-interference ability of the sensor. The response current of the sensor to other anions at a 10-fold concentration was less than 4%, indicating that the sensor still exhibited good selectivity within the detection range of high-concentration nitrates.

The sensor exhibited good repeatability and durability. The relative standard deviation (RSD) of the current responses during 10 consecutive tests in 1.5 mg/L nitrate standard solutions was 0.92%. The RSD of the current responses was only 1.03% for 25 consecutive tests in a 10 mg/L nitrate standard solution. The sensor’s RSD was 2.02% after a 7-day assay using a standard solution with a concentration of 5 mg/L. The modified electrode was tested for over 300 LSVs, demonstrating consistent electrocatalytic performance. The detection sensitivity was 5.59 µA·mg·L^−1^ after 300 tests. Due to the high chemical stability and hardness of BDD, the sensor is highly corrosion-resistant and robust. 

Table 1 shows the performances of the developed sensor and other reported electrochemical nitrate sensors. In comparison with other reported sensors, the nitrate sensor developed in this study has a wider detection range, with an upper limit of 100 mg/L (7143 μM). In addition, the sensor offers high stability compared to other sensors. The copper modification method for the BDD electrode described in this study involved simple electrodeposition. This, combined with the stability of BDD, resulted in better reproducibility of the sensing electrode compared to other nitrate-sensing electrodes that require laborious preparation and modification processes. The RSD of the current responses was only 1.03% for 25 consecutive tests in a 10 mg/L nitrate standard solution. The wide detection range and high stability of the sensor makes it suitable for field applications and the on-site monitoring of nitrate-contaminated waters. Moreover, the BDD electrode surface can be renewed by applying a positive voltage to clean the electrode surface of modified copper and other contaminants, and the electrode can be re-modified and reused in other assays. This allows for the use of a single BDD electrode for extended periods at high frequencies and further extends the lifespan of the sensor.

### 3.6. Determination in Actual Water Samples

The sensor was used to detect the concentration of nitrate in actual water samples. The actual water samples were selected from local lake water. H_2_SO_4_ and electrolytes were added to the actual water samples to adjust them to a suitable electrolyte environment before electrochemical testing. For comparison, ion chromatography was also used for the determination of nitrate in the lake water samples. The detection results using the developed sensor and ion chromatography are shown in Table 2. In comparison with the detection results of ion chromatography, the recovery obtained with the developed sensor was between 87.8% and 112.4%. The test results indicated that the electrochemical sensor based on the Cu-modified BDD electrode prepared in this study has the potential to be applied to the detection of nitrate in real water samples.

## 4. Conclusions

In this study, the catalytic effect of Cu on nitrate was used to construct a nitrate detection sensor based on a customized BDD electrode. Through optimization of the concentration and pH of the electrolyte solution, the detection range of the sensor was broadened to 0.07~100 mg/L, enabling the detection of nitrate at high concentrations. The detection sensitivity of the high concentration band was 5.33 µA·mg·L^−1^, and the detection limit was 0.065 mg/L. The relative standard deviation (RSD) of the current responses during 10 consecutive tests in 1.5 mg/L nitrate standard solutions was 0.92%. The RSD of the current responses was only 1.03% for 25 consecutive tests in a 10 mg/L nitrate standard solution. This nitrate sensor exhibited excellent anti-interference properties and selectivity and high stability, which makes it a promising candidate for the on-site monitoring of nitrate-polluted water bodies.

## Figures and Tables

**Figure 1 micromachines-15-00487-f001:**
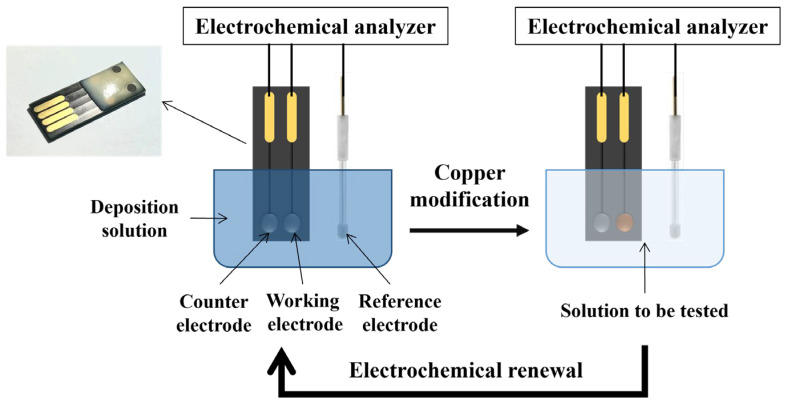
Nitrate electrochemical sensor modification, detection, and electrochemical renewal process.

**Figure 2 micromachines-15-00487-f002:**
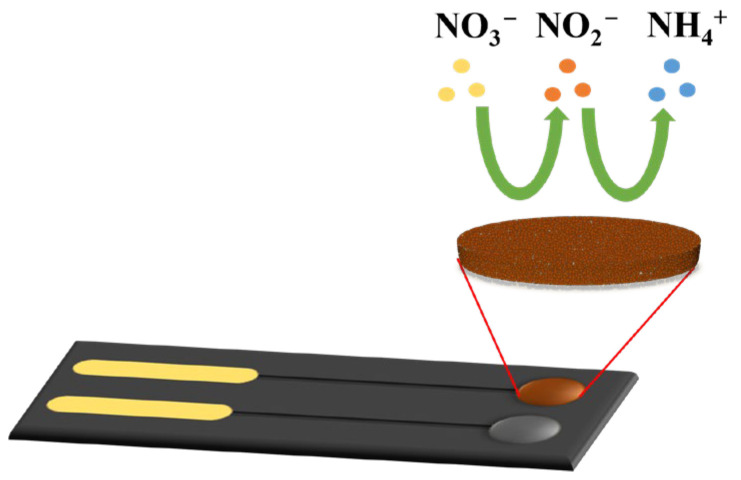
Nitrate detection process using the electrochemical sensor based on a BDD electrode.

**Figure 3 micromachines-15-00487-f003:**
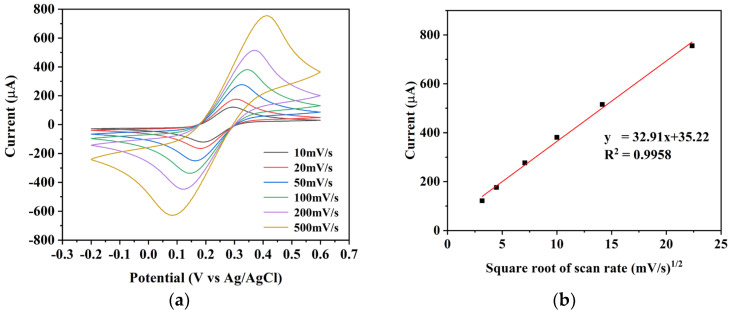
(**a**) Cyclic voltammetry curves of the BDD electrode in a 5 mM Fe(CN)_6_^3−/4−^ potassium ferricyanide system at different scanning speeds. (**b**) Plot of the square root of peak current versus sweep rate at different sweep rates.

**Figure 4 micromachines-15-00487-f004:**
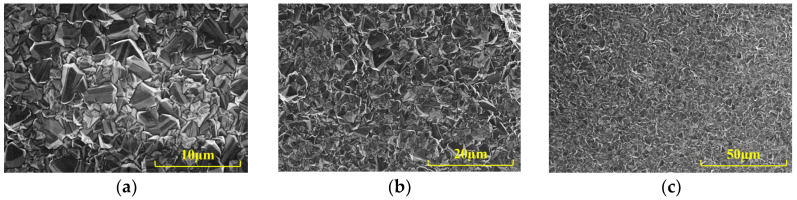
SEM images of the bare BDD electrode at (**a**) ×5000, (**b**) ×2500, and (**c**) ×1000 magnification, and the BDD electrode with copper modification at (**d**) ×5000, (**e**) ×2500, and (**f**) ×1000 magnification.

**Figure 5 micromachines-15-00487-f005:**
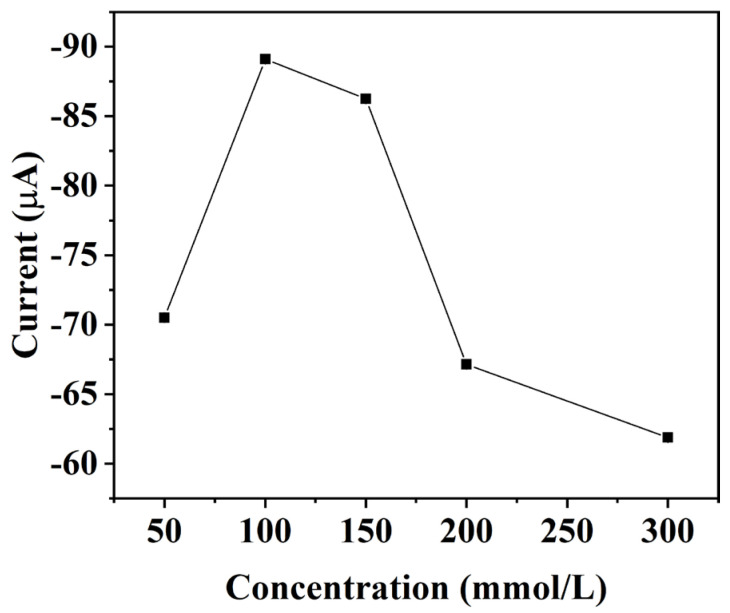
Influence of Na_2_SO_4_ concentration on the reduction current of nitrate.

**Figure 6 micromachines-15-00487-f006:**
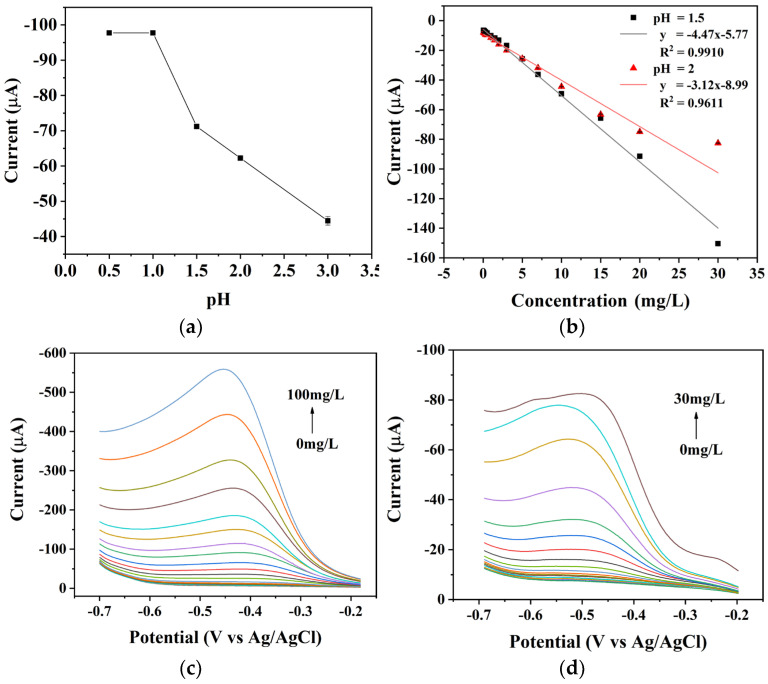
(**a**) Influence of substrate pH on the reduction current of nitrate. (**b**) Linear scanning voltammograms of the BDD electrode measured in nitrate solutions at different concentrations (0–100 mg/L) in substrate pH = 1.5. (**c**) Linear scanning voltammograms of the BDD electrode measured in nitrate solutions at different concentrations (0–30 mg/L) in substrate pH = 2. (**d**) Calibration curves for nitrate with substrate pH = 1.5 and 2.

**Figure 7 micromachines-15-00487-f007:**
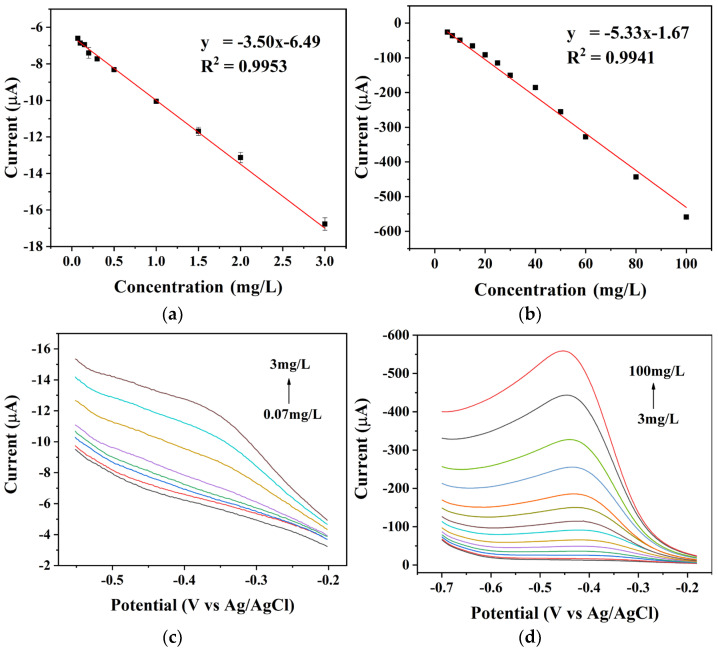
Calibration curves for nitrate at concentrations of (**a**) 0.07–3 mg/L and (**b**) 3–100 mg/L. Linear scanning voltammograms of the BDD electrode measured in nitrate solutions at different concentrations in the ranges of (**c**) 0.07–3 mg/L and (**d**) 3–100 mg/L.

**Figure 8 micromachines-15-00487-f008:**
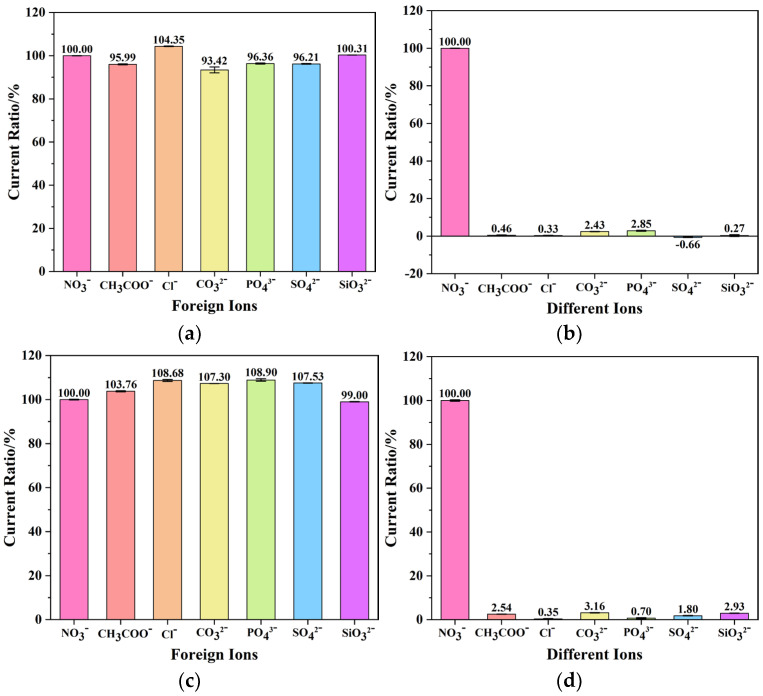
(**a**) Interference test results and (**b**) selectivity test results at a nitrate concentration of 1 mg/L in the presence of interfering ions at a concentration of 10 mg/L. (**c**) Interference immunity test results and (**d**) selectivity test results at a nitrate concentration of 10 mg/L and in the presence of interfering ions at a concentration of 100 mg/L.

**Table 1 micromachines-15-00487-t001:** Performance comparison of different nitrate electrochemical sensors.

Electrode	Method	LOD (nM)	Linear Range (μM)	Sensitivity	RSD	Ref.
Ag nanoparticles/Au electrode	SWV	0.9	0.0009–1000	12 µA mM^−1^	-	[32]
Cu/Pt microelectrode	CA	-	0–3500	31 µA mM^−1^	2.48% n = 15	[33]
Pt/ZnO/chitosan	CV	10	100–2000	39.91 μA/cm^2^ mM	0.52% n = 10	[34]
Cu microspheres/polyaniline/micro-needle	DPV	8000	20–6000	141.69 μA/cm^2^ mM	2.8% n = 50	[35]
Cu-SPCEs	LSV	91	50–750	0.1042 μA m M^−1^	0.5% n = 5	[36]
Copper nanowire array	CV	9000	10–1500	0.73 μA μM^−1^ cm^−2^	7% n = 3	[23]
Cu/BDD electrode	LSV	4600	5–214 214–7143	0.25µA mM^−1^ 0.38µA mM^−1^	1.03% n = 25	This work

**Table 2 micromachines-15-00487-t002:** Analysis results of nitrate in actual water samples.

Sample	Determined by Ion Chromatography (mg/L)	Determined by This Sensor (mg/L)	Recovery (%, n = 3)
Lake water 1	1.37	1.54	112.4
Lake water 2	5.02	4.54	90.4
Lake water 3	5.32	4.67	87.8

## Data Availability

The data that support the findings of this study are available from the corresponding authors upon reasonable request.

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
