# Peer review of "Nitrate Sensor with a Wide Detection Range and High Stability Based on a Cu-Modified Boron-Doped Diamond Electrode"

_micromachines, 2024, doi:10.3390/mi15040487_

Round 1
Reviewer 1 Report
Comments and Suggestions for Authors
The authors present a Cu-modified boron-doped diamond (BDD) electrode used as an electrochemical sensor for nitrate detection in waters. By optimizing the electrolyte system, the linear detection range was expanded allowing the sensor to detect concentrated nitrate samples up to 100 mg/L, which makes it suitable for field applications and on-site monitoring of nitrate-contaminated waters. But the authors need to supply more information to demonstrate that the stability of the sensor was improved. Thus, the manuscript can be published after providing supplementary information on the stability of the sensor.
Reviewer 2 Report
Comments and Suggestions for Authors
This paper demonstrates an electrochemical sensor based on Cu-modified boron-doped diamond (BDD) electrodes for the detection of nitrate-contaminated water.
The sensor takes advantage of copper's catalytic effect on nitrate and the stability of the BDD electrode. By optimizing the electrolyte system, the linear detection range is expanded, enabling the sensor to detect high-concentration nitrate samples up to 100 mg/L with a detection limit as low as 0.065 mg/L. The sensor has a wide detection range and high stability, making it suitable for field applications and on-site monitoring of nitrate-polluted water bodies.
I think this paper designed a high-quality electrochemical sensor that will be of great use for future applications. It is recommended to post after improving the comments below.
1. What is the relationship between the detection concentration and the electrochemical sensor?
2. How accurate is the detection of multiple electrochemical sensors?
3. Is there a huge change in the accuracy of repeated testing and testing a few days later?
